# Cardiorespiratory Effects and Desflurane Requirement in Dogs Undergoing Ovariectomy after Administration Maropitant or Methadone

**DOI:** 10.3390/ani13142388

**Published:** 2023-07-23

**Authors:** Francesca Cubeddu, Gerolamo Masala, Giovanni Sotgiu, Alessandra Mollica, Sylvia Versace, Giovanni Mario Careddu

**Affiliations:** 1Department of Veterinary Medicine, University of Sassari, Via Vienna 2, 07100 Sassari, Italy; gemasala@uniss.it (G.M.); alessandra.mollica74ss@gmail.com (A.M.); versace.sylvia@gmail.com (S.V.); gcareddu@uniss.it (G.M.C.); 2Department of Surgical and Experimental Medical Sciences Medicine, Surgery and Pharmacy, University of Sassari, Viale San Pietro 43, 07100 Sassari, Italy; gsotgiu@uniss.it

**Keywords:** general anesthesia, dog, ovariectomy, maropitant, methadone, desflurane requirement, cardiorespiratory functions, pain management

## Abstract

**Simple Summary:**

Opioids such as methadone are the most potent and most used analgesic drugs in anesthetic protocols, but they have several dose-dependent adverse effects. Some drugs other than opioids also have analgesic effects. Analgesic drugs in the anesthetic protocol can reduce the requirement of other drugs, particularly inhalant agents. Maropitant is an antiemetic for dogs and cats that has been shown to also exert analgesic effects, especially on visceral pain. The purpose of this study is to evaluate the cardiorespiratory effects and analgesic properties of maropitant and methadone combined with desflurane in dogs undergoing ovariectomy. Forty dogs were randomly assigned to receive either maropitant or methadone. Maropitant produced analgesia and reduced the requirement of inhalant agent in amounts very similar to those determined by methadone, while maintaining heart rate, arterial blood pressure, respiratory rate and carbon dioxide end-tidal partial pressure even at a more satisfactory level. Therefore, maropitant can be suggested as an analgesic drug for abdominal surgery not only in healthy dogs but also in those with reduced cardiorespiratory compensatory capacities or at risk of hypotension, especially when combined with a sedative such as dexmedetomidine.

**Abstract:**

General anesthesia for ovariectomy in dogs is based on a balanced anesthesia protocol such as using analgesics along with an inhalant agent. While opioids such as fentanyl and methadone are commonly used for their analgesic potency, other drugs can also have analgesic effects. Maropitant, an antiemetic for dogs and cats, has also been shown to exert analgesic effects, especially on visceral pain. The aim of this study was to compare the cardiorespiratory effects and analgesic properties of maropitant and methadone combined with desflurane in dogs undergoing ovariectomy. Two groups of 20 healthy mixed-breeds bitches undergoing elective ovariectomy received intravenous either maropitant at antiemetic dose of 1 mg kg^−1^ or methadone at the dose of 0.3 mg kg^−1^. Cardiorespiratory variables were collected before premedication, 10 min after sedation and during surgery. Recovery quality and postoperative pain were evaluated 15, 30, 60, 120, 240 and 360 min postoperatively. Results showed that maropitant produced analgesia and reduced the requirement of desflurane in amounts similar to those determined by methadone (5.39 ± 0.20% and 4.91 ± 0.26%, respectively) without significant difference, while maintaining heart rate, arterial blood pressure, respiratory rate and carbon dioxide end-tidal partial pressure even at a more satisfactory level. Therefore, maropitant may be recommended as an analgesic drug for abdominal surgery not only in healthy dogs but also in those with reduced cardiorespiratory compensatory capacities or at risk of hypotension, especially when combined with a sedative such as dexmedetomidine.

## 1. Introduction

General anesthesia for ovariectomy surgery in dogs is based on a balanced anesthesia protocol which, in addition to an inhalant agent that causes unconsciousness and amnesia, also includes drugs with analgesic effect. Among the analgesics used for this purpose, opioids stand out for their potency [1,2,3]. In current veterinary practice, the use of analgesic drug combinations acting on different sites and pathways, “perioperative multimodal analgesia”, is often advocated to maximize analgesic effect. In surgical patients, the multimodal and preemptive approach for analgesia is commonly achieved by combining opioids with non-steroidal anti-inflammatory drugs [4,5]. The use of opioids is often restricted due to legal issues [6,7], poor knowledge of opioid pharmacology [7] and their adverse effects [8,9,10]. Because of these adverse effects, the use of other drugs with opioid-like analgesic properties is a research field in companion animal anesthesia [11].

Maropitant is a neurokinin 1 (NK-1) receptor antagonist approved for use as an antiemetic for dogs and cats [12,13,14,15] that has also been demonstrated to exert an analgesic effect on abdominal pain in a number of animal studies [16,17,18,19,20,21,22,23,24]. Maropitant provides antiemetic effects by preventing the substance P (SP) from binding with the NK-1 receptors located in the vomiting center and the chemo-trigger zone [5]. The highly selected NK-1 receptor antagonism of maropitant that blocks the effect of SP on the central nervous system is therefore expected to provide analgesic effects through the same mechanism at the dorsal horn level [5]. Although its mechanism of action is very different from that of opioids and not yet fully understood, some studies encourage its use in combination with other analgesic agents within the anesthesia protocol [22,25]. Maropitant was shown to decrease halogenated anesthetic requirements in dogs where noxious stimulation to the ovary and ovarian ligament was applied [22,26]. At an antiemetic dose of 1 mg kg^−1^, maropitant decreased the sevoflurane minimum alveolar concentration (MAC) by 24% in dogs and 15% in cats [25].

A transient decrease in mean arterial pressure has been observed lasting approximately 10 min when maropitant was administered in healthy dogs under general anesthesia [22,27]. This effect may be due to its NK-1 receptor antagonism which occurs both centrally and peripherally, producing a transient decrease in arterial pressure and an increase in the reflex heart rate in dogs. The underlying mechanism is unknown but may be associated with modulation of both the cardiovascular system and the sympathetic system by central NK-1 receptors [28].

Dexmedetomidine is an *α*2-adrenoceptor agonist commonly administered in dogs for premedication to provide dose-dependent sedation, analgesia and muscle relaxation [29]. Alpha2-adrenoceptor agonists reduce the requirements for other anesthetics and their analgesic and sedative effects are enhanced when combined even at low doses with opioids [30,31]. The increase in systemic vascular resistance exerted by dexmedetomidine is higher in dogs anesthetized with desflurane than in dogs anesthetized with isoflurane [32].

Methadone is a pure opioid agonist of synthetic origin, with triple analgesic potency compared to morphine [33]. Its analgesic effect is mainly due to full agonism at μ-opioid receptors and non-competitive antagonism at N-methyl-D-aspartate (NMDA) receptors. Dose-dependent adverse effects of methadone include a reduction in heart rate caused by an increase in vagal tone; dysphoria; restlessness polypnea with shallow breathing; and respiratory depression [33,34]. Methadone can be administered via intravenous (IV) or intramuscular (IM) route alone or in combination with sedative agents such as α2-agonists in premedication, or even intraoperatively. Its IV dosage in dogs is 0.25–0.50 mg kg^−1^ [33,35].

Desflurane is highly fluorinated and ether-derived and was recently introduced in veterinary clinical practice. In dogs, it has a low blood/gas partition coefficient (0.63) and an anesthetic potency of about 1/3 compared to sevoflurane [36,37,38]. The poor lipid solubility of desflurane results in low potency and, consequently, high MAC (7.2%) [38,39]. In humans, MAC for desflurane has been shown to be very variable (3–10%) and strongly influenced by factors such as age, administered premedication and body temperature. In dogs, Hammond (1994) found desflurane MAC to be 10.32 ± 0.14%; when 50% nitrous oxide was added to the carrier gas, it was reduced to 7.99 ± 0.57% [39]. Unlike other inhalant agents, desflurane administered at high concentrations in the first minutes of anesthesia maintenance causes a sympathetic–mimetic stimulation resulting in tachycardia, hypertension and increased vasopressin and adrenaline levels in the blood [37,40,41,42].

The aim of this study was to compare desflurane requirement, heart rate, arterial pressure, respiratory rate, carbon dioxide end-tidal partial pressure and analgesic effects when maropitant or methadone was administered within a balanced anesthesia protocol with dexmedetomidine, propofol and desflurane in dogs undergoing ovariectomy.

Our hypothesis was that maropitant at a dose of 1 mg kg^−1^ [25] lowers the requirement for desflurane and reduces cardiorespiratory functions to amounts similar to those determined by methadone at a dose of 0.3 mg kg^−1^ [31]; both of these drugs are administered intravenously as analgesic premedication agents in canine ovariectomy.

## 2. Materials and Methods

### 2.1. Ethical Statement

The study was approved by OPBSA Ethical Committee of the University of Sassari under Protocol No. 1820/2017. Dog owners received information about the anesthesia and the procedure and authorized the inclusion of animals in the study by signing an informed consent form.

### 2.2. Study Subjects

Using randomization software (https://stattrek.com/statistics/random-number-generator, accessed on 21 December 2020), 40 healthy mixed-breeds bitches undergoing elective ovariectomy were assigned to one of two groups (20 subjects each) to receive one of two general anesthesia protocols that differed only in maropitant or methadone inclusion; these groups were indicated as the Maropi group and the Metha group, respectively.

Only dogs aged 9–24 months, weighing 15–25 kg, free from any clinically evident cardiorespiratory abnormality, hematocrit or total serum protein alteration, heat or pregnancy in progress or already occurred and assigned an ASA I score (American Society of Anesthesiologist) were included in the study. The dogs were fasted for 12 h and deprived of water for 6 h before the clinical trial.

### 2.3. Anesthetic and Analgesic Protocol/Management

For premedication, dexmedetomidine 5 μg kg^−1^ (Dextroquillan^®^ 0.05% A.T.I. S.r.l., Bologna, Italy) and maropitant 1 mg kg^−1^ (Vetemex^®^ 1% Virbac S.r.l., Milano, Italy) were prepared in the same syringe for the dogs in the Maropi group, while dexmedetomidine 5 μg kg^−1^ and methadone 0.3 mg kg^−1^ (Semfortan^®^ 1% Dechra Veterinary Products Srl, Torino, Italy) were prepared for the dogs in the Metha group by an operator who was the only person aware of the contents of the syringes and was not involved in the study. Premedication was diluted to 5 mL with saline solution for both groups and was IV injected through left cephalic vein access, always by the same veterinarian, while ringer lactate solution IV 5 mL kg^−1^ h^−1^ (Ringer Lattato S.A.L.F.^®^ Bergamo, Italy) started to be administered. When sedation was achieved, left metatarsal artery access was obtained for invasive systolic (SAP), diastolic (DAP) and mean (MAP) arterial blood pressure measurement. Glucose 2.5% in ringer lactate was available for IV administration in dogs with glycemia < 70 mg dL^−1^.

Induction was achieved with propofol IV (Proposure^®^ 1%, Merial Italia, S.p.a., Milano, Italy) at the effective dose for successful intubation and the endotracheal tube was connected to the pediatric circle system of the workstation for inhalant anesthesia (Fabius GS, Dräger^®^, Dräger Medical Italia S.p.A., Milano, Italy) with the dog in spontaneous ventilation. Desflurane (Suprane^®^ Baxter Italia, Firenze, Italy) in oxygen (O_2_)/air was delivered through an agent-specific out-of-circuit vaporizer with fresh gas flow set to 1.0 L min^−1^ to maintain O_2_ inspiratory fraction at 0.35–0.40. Calibration of the gas module (Scio four Oxi Plus, Dräger^®^) was verified daily (Quick Calibration Gas^®^, Datex-Ohmeda, GE HealthCare, Milano, Italy, 2.0% desflurane, 5% carbon dioxide, 54.5% oxygen, 36% nitrous oxide and 2.5% nitrogen). Based on the results previously obtained during 8 pilot studies, a 20 min time with desflurane end-tidal percentage (EtDes) at 5.5% was elapsed for inhalant anesthetic equilibration before the start of surgery. The EtDes was adjusted and maintained at a level sufficient to ensure a surgical plane of anesthesia, verified by the absence of palpebral reflex, jaw tone, abdominal straining and MAP between 60 and 100 mmHg. In case of positive response to each painful stimulus, desflurane delivery was increased by 1% while otherwise reduced by 0.5%. End-tidal percentages of desflurane were converted into MAC multiples considering 7.64% as the MAC in the dog [43].

Fentanyl (Fentadon^®^ 0.5%, Dechra Veterinary Products Srl, Torino, Italy) 0.005 mg kg^−1^ was ready to be IV administered as rescue analgesia should the animal respond with movement or otherwise to surgical stimulation, i.e., a more than 20% increase in either respiratory rate (RR), heart rate (HR) and/or MAP. Norepinephrine (Noradrenalina^®^ 0.1%, S.A.L.F., Bergamo, Italy) was available for IV administration 0.1–2 μg kg^−1^ min^−1^ after 5 min of MAP < 65 mmHg and/or SAP < 90 mmHg [37].

The dogs were left on spontaneous ventilation with the option of performing volume-controlled mechanical ventilation after 15 s of apnea. Desflurane administration was discontinued at the end of the surgery; the dogs breathed 100% oxygen and the extubation time was recorded.

### 2.4. Surgical Procedure

Ovariectomy was always performed by the same team through a 4–5 cm median celiotomy just caudal to the umbilical scar with the dog positioned in dorsal recumbency onto a resistive heating mat.

Traction force of 1 min duration was always exerted by the same surgeon, first on the right and then on the left ovarian ligament, sufficient for ovarian pedicle ligation and subsequent ovariectomy. Traction force was measured three times in kg load using a handheld digital dynamometer (Kop 24382, Keen Optics, Heyuan, China) calibrated with Ohaus hook weights (Ohaus international); average values were then converted into N [3].

### 2.5. Data Collection and Evaluation Times

All variables were always detected by the same veterinarian, as reported in Table 1.

Heart rate, RR and rectal temperature (RT) were recorded shortly before intravenous access as a baseline measurement (T0), ten minutes after premedication (T1), just before induction (T2) and then at each time point until the end of the surgery (T3–T11).

Sedation scores were recorded at T1 according to the composite simple descriptive sedation score for dogs, as described in the work of Grint 2009 and Wagner 2017 (Table 2) [43,44].

Intubation scores were recorded at T3 as described in Table 3.

Times for sedation achievement and for successful intubation were also recorded.

Tidal volume (VT), minute respiratory volume (VM), carbon dioxide end-tidal partial pressure (PetCO_2_), arterial oxygen saturation (SpO_2_), EtDes, SAP, DAP and MAP began to be recorded from immediately after intubation (T3) until skin suture (T11) as they were displayed on the monitor (Infinity Delta, Dräger^®^).

Values of right and left ovarian ligament traction forces were recorded at T6 and T8, respectively.

### 2.6. Recovery Quality and Postoperative Analgesic Management

Recovery quality was evaluated using the scoring system (Table 4) described by Hampton 2019 [45].

Any postoperative pain manifestations were always objectively assessed by the same experienced veterinarian, who was unaware of the dog’s grouping, at the subsequent postoperative times: 15, 30, 60, 120, 240 and 360 min, using the validated Italian translation (Della Rocca 2018) of the Glasgow Composite Measure Pain Scale-Short Form (CMPS-SF) (Table 5) [46,47].

When necessary, postoperative rescue analgesia was available to be provided by methadone IV 0.2 mg kg^−1^.

Meloxicam (Metacam^®^ 0.5%, Boehringer Ingelheim Animal Health Italia Spa, Padova, Italy) IV 0.2 mg kg^−1^ was administered in dogs of both groups after the last round of data collection, six hours after the end of surgery.

The same veterinarian evaluated head lifting, sternal recumbency and standing/walking times.

### 2.7. Statistical Analysis

A priori power analysis was conducted using G*Power 3.1.9.7 software to determine the sample size needed to achieve adequate statistical power (>85%). All data were tested for normal distribution via the Shapiro–Wilk test. Results are expressed as means ± standard deviations (SD) or as medians and interquartile ranges (IQRs). Analysis was performed using the Stata 17 software (StataCorp, College Station, TX, USA). The results were considered significantly different for *p* < 0.05.

## 3. Results

The collected data were confirmed for normality of distribution. Dogs were 16.95 ± 4.17 months of age in the Maropi group and 15.90 ± 4.19 months in the Metha group, while the body weight was 20.85 ± 2.85 kg in the Maropi group and 19.95 ± 2.31 kg in the Metha group, without significant difference between groups. No significant differences were found for hematocrit and for total serum protein concentration.

The mean duration of the surgery was 46 ± 7 min, SpO_2_ was always above 95%, RT never dropped below 37.2 °C and the amount of propofol administered was 2.2 ± 0.6 mg kg^−1^, without significant differences between groups.

No need for glucose administration occurred in any dog.

No significant differences were found between the two groups for sedation, intubation and recovery scores expressed as median (IQRs), i.e., 14 (14–14.25), 0 (0–0.25), 0.5 (0–1) and 14 (14–14), 0 (0–0), 0 (0–1) for the Maropi group and the Metha group, respectively.

No significant differences were found between the two groups for postoperative pain scores at recovery and at 15, 30, 60, 120, 240 and 360 min after recovery, expressed as median (IQRs), i.e., 0 (0–1), 0 (0–1), 0 (0–0.25), 0 (0–0), 0 (0–0) and 0 (0–1), 0 (0–1), 0 (0–1), 0 (0–0), 0 (0–0) for the Maropi group and the Metha group, respectively.

The traction force exerted on the ovarian ligaments representing peak noxious stimulation was 7.8 ± 0.9 and 7.3 ± 0.7 N for the right and 6.3 ± 1.0 and 6.8 ± 0.9 N for the left ovary in the Maropi group and the Metha group, respectively, without significant difference.

All detected cardiovascular variables (HR, SAP, DAP and MAP) significantly increased in both groups at ovarian ligaments traction (T6, T8) compared with the same variables shortly before surgery (T4), reaching the highest values of the entire trial in each dog of both groups and remaining within normal ranges. Heart rate was significantly higher in the Maropi group compared with the Metha group from T1 to T11 (Figure 1); SAP, DAP and multiples of MAP were also significantly higher in the Maropi group compared with the Metha group from T3 to T11 (Figure 2, Figure 3 and Figure 4). No need for norepinephrine administration occurred.

The same pattern was shown by RR, which was significantly higher in the Maropi group than in the Metha group ten minutes after administration of premedication (T1) and then from immediately after intubation (T3) until skin suture (T11) (Figure 5).

Simultaneously with higher RR, PetCO_2_ showed significantly lower values in the Maropi group; however, values were maintained within the normal range, never dropping below 38 mmHg. Only immediately after intubation (T3) and at fascia suture (T10), did PetCO_2_ reach values higher than 45 mmHg, suggestive of mild hypercapnia, in the Metha group (Figure 6). Ventilatory assistance was never required since the rare cases of bradypnea were of short duration and apnea never occurred.

The tidal volume was always slightly higher in the Metha group, while VM was always higher in the Maropi group, both without statistical significance (Table 6).

The EtDes at each detection time during surgery (T5–T11) was higher in the Maropi group than in the Metha group without statistical significance (Figure 7).

The average of EtDes of the entire surgery expressed in MAC multiples was 0.70 ± 0.03 in the Maropi group and 0.64 ± 0.03 in the Metha group when referred to 7.64% as MAC in the dog [48] without statistical significance (Figure 8).

No need for rescue analgesia occurred in any dog in both groups during surgery.

No need for rescue analgesia occurred in any dog in both groups postoperatively, since average ± SD postoperative pain scores (CMPS-SF) were 0.40/20 ± 0.50, 0.30/20 ± 0.47, 0.25/24 ± 0.44, 0.15/24 ± 0.37, 0.00/24 ± 0.00 and 0.45/20 ± 0.51, 0.35/20 ± 0.49, 0.30/24 ± 0.47, 0.20/24 ± 0.41, 0.00/24 ± 0.00 for Maropi group and Metha group, respectively.

Sedation time was shorter in dogs in the Maropi group without statistical significance, while extubation, head lifting, sternal recumbency and standing/walking times were shorter with statistical significance (Figure 9).

## 4. Discussion

The work we conducted is a blinded clinical study for a routine, standardized and short-term surgery such as dog ovariectomy.

Significantly higher ventilatory and cardiovascular variables throughout the procedure in dogs receiving maropitant are indicative of less cardiorespiratory depression than in dogs receiving methadone, in which the greater cardiorespiratory depression is presumably due to increased vagal tone resulting from full agonism of μ-opioid receptors and non-competitive antagonism of N-methyl-D-aspartate (NMDA) receptors [33,34]. Unlike opioids, the mechanism of action of maropitant is associated with the modulation of the cardiovascular system and the sympathetic system by central NK-1 receptors [49].

Sedation and intubation times were similar in the two groups, and intubation occurred on the first attempt and at similar times in all dogs in both groups, with the same propofol dosages and showing the best score of the scale, demonstrating that maropitant is a component just as valid as methadone for premedication with dexmedetomidine, at the doses here proposed.

The shorter extubation, head lifting, sternal recumbency and standing/walking times in the Maropi group could be attributable to a shorter half-life and shorter action of maropitant compared to methadone [24,33].

The good and comparable recovery quality and postoperative pain scores assessed in both groups demonstrate that in an abdominal surgery, maropitant can abolish postoperative pain similarly to an opioid such as methadone.

The rapid recovery time has presumably also been favored by the maintenance of RT above 37 °C by the heating mat, as washout from inhalant agents is slowed down by low temperature [39,50].

The significant increase in cardiovascular variables at ovarian ligaments traction (T6 and T8) compared to preoperative values (T4), shows that these two maneuvers were the strongest noxious stimulations and are presumably related to the surgical technique adopted. Nevertheless, rescue analgesia was never required in any dog from either group.

The close similarity between the two groups in the increase in cardiovascular variables at ovarian ligaments traction (T6 and T8) compared to preoperative values (T4), presumably indicates that the analgesic effect of methadone and maropitant at the dose here employed are very comparable.

The average of traction forces exerted in this study at ovarian ligament ligation were very close to those exerted by Columbano 2012 [3] in the group in which fentanyl was administered (7.5 vs. 7.7 N for the right and 6.5 vs. 6.7 for the left ovarian ligaments, respectively).

Desflurane was chosen as an inhalant anesthetic; compared to other agents, it currently leads to the fastest uptake and washout, so that the depth of anesthesia can be managed more quickly based on any pain responses [36,38,51,52].

It was decided to leave the patients in spontaneous ventilation to better evaluate possible influences of the administered anesthetic agents on the respiratory activity. The maintenance of RR, VT, VM and PetCO_2_ within normal ranges in all dogs is indicative of their good cardiorespiratory conditions throughout the trial.

Dexmedetomidine, even at the low dose of 5 μg kg^−1^, also guaranteed the presence of an agent with analgesic effect in the dogs in the Maropi group that did not receive any opioid.

It was decided not to administer vasoactive drugs until the MAP decreased below 65 mmHg for at least 5 min. As there are currently no veterinary guidelines to support a given MAP cutoff value, the 65 mmHg limit was chosen in accordance with the survey results of Murphy 2022 [53]. The non-administration of vasoactive agents enhances the reliability of cardiovascular data and their comparison between the two groups.

In our work, maropitant was administered at the same dose as in the study of Marquez 2005 [25] (1 mg kg^−1^) who showed it to have a greater analgesic effect than morphine at the dose of 0.5 mg kg^−1^ (1.06 MAC and 1.17 MAC, respectively).

In the latter study, isoflurane was administered instead of desflurane, without any α2-agonist, and ovariohysterectomy was performed instead of ovariectomy, a surgery that was slightly more invasive and longer. During surgical stimulation, the isoflurane requirement was 1.36% with maropitant and 1.51% with morphine, i.e., 1.06 MAC and 1.17 MAC, respectively, when referred to 1.28% as isoflurane MAC in dogs [29,54,55].

With the anesthetic protocols we employed for an abdominal surgery such as ovariectomy in the dog, maropitant 1.0 mg kg^−1^ and methadone 0.3 mg kg^−1^ reduce the desflurane requirement to 0.70 MAC and to 0.64 MAC, respectively, when referred to 7.64% as desflurane MAC in dogs [48].

The response in sevoflurane requirement observed by Boscan 2011 [26] in the dog model during ovary and ovarian ligament stimulation was similar to the clinical response observed during visceral traction in anesthetized dogs.

The lower MAC multiples obtained in our work both in dogs with maropitant (0.70 MAC) and in dogs with methadone (0.64 MAC) compared to those obtained by Marquez 2015 [25] (1.06 MAC with maropitant and 1.17 MAC with morphine) are presumably due to the presence of dexmedetomidine, even at the low dose of 5 µg kg^−1^ of our study.

The small difference in desflurane requirement between the two groups suggests that maropitant at the dose of 1.0 mg kg^−1^ has only slightly less analgesic effect than methadone at a dose of 0.3 mg kg^−1^.

Boscan 2011 [22] observed a reduction in the inhalant agent requirement to 0.76 MAC in ovarian traction stimulation in dogs when maropitant was administered at the same dosage as ours (1 mg kg^−1^), while a reduction to 0.70 MAC was obtained only with a five times higher dose (5 mg kg^−1^), referring to their sevoflurane MAC-BAR of 2.12%. This lower reduction in inhalant agent requirement compared with ours is presumably because inhalant agent was the only anesthetic administered.

In dogs tested with 50 V, 50 Hz, 10 ms on the upper gingival as noxious stimulus, Fukui 2017 [5] observed a reduction in the sevoflurane requirement to 0.85 MAC with maropitant 1 mg kg^−1^ and to 0.83 MAC with the combination of maropitant 1 mg kg^−1^ and carprofen 4 mg kg^−1^. Also in this comparison, the lower inhalant agent requirement found in our work (0.70 MAC) was presumably advantaged by the presence of dexmedetomidine.

A slight decrease in arterial blood pressure with a concomitant increase in HR was observed by Chi T-T 2020 [28] after administration of maropitant 1 mg kg^−1^ in awake dogs. In the same work, maropitant 1 mg kg^−1^ produced clinically significant hypotension in dogs premedicated with acepromazine 5 µg kg^−1^, induced with propofol and maintained on isoflurane, while the same dogs premedicated with dexmedetomidine 5 µg kg^−1^ did not experience any significant decrease in arterial blood pressure. Depression of compensatory response to hypotension during maropitant injection could be due to inhibition of sympathetic activity, adrenergic neurotransmission, and baroreceptor reflex sensitivity exerted by halogenated and exacerbated by phenothiazine derivate [28,56]. Presumably, it is preferable that when maropitant is present in a multidrug premedication, it is combined with an α2-agonist such as dexmedetomidine rather than a phenothiazine such as acepromazine.

Such an arterial pressure reduction after maropitant administration may also be an effect of sulfobutylether-beta-cyclodextrin and metacresol contained within the formulation of injectable maropitant used (Cerenia^®^, Zoetis Inc, Kalamazoo, MI, USA) as in other formulations [28,57,58,59,60,61].

The absence of the two above-mentioned excipients in the formulation of injectable maropitant (Vetemex^®^ 1%, CP-Pharma, Handelsgesellschaft mbH, Ostlandring 13, Burgdorf, Germany) used in our study might be the reason for the absence of a significant drop in arterial blood pressure in the dogs in the Maropi group. Furthermore, the premedication containing alfa 2-agonist and no phenothiazine may be the reason for maintaining arterial blood pressure values in the upper-medium range [56].

The slight increase in cardiovascular and respiratory variables observed at skin suture (T11) in the Maropi group reflects the lower somatic analgesic effect of maropitant if compared with methadone [22,25,26].

The properties of maropitant, combined with desflurane, could provide an acceptable analgesia protocol comparable to methadone and also could help to reduce patient recovery times and the consequences that a prolonged recovery might represent for dogs. Smooth recovery and faster return to eating may allow improved overall patient recovery, earlier discharge and increased owner satisfaction [62].

## 5. Conclusions

Maropitant lowers the requirement for desflurane in amounts very similar to those determined by methadone, maintaining cardiorespiratory functions even at a more satisfactory level.

These results encourage the inclusion of maropitant as an analgesic drug in anesthetic protocols for abdominal surgery not only in healthy dogs, but also in those with reduced cardiorespiratory compensatory capacities or at risk of hypotension (polytrauma, pyometra, cesarean section, liver or kidney failure) or in patients with reduced cardiorespiratory compensatory capacities (pediatric and geriatric patients), especially when combined with a sedative such as dexmedetomidine.

A co-induction with propofol and ketamine (Ketofol) [63] after the premedication used in our Maropi group (dexmedetomidine and maropitant) could result in an anesthetic protocol that is also appropriate for somatic surgery, given the somatic analgesic effect of ketamine [64,65,66]. Adding ketamine to propofol and subsequently reducing their doses could greatly reduce the side effects of both agents.

The longer half-life of maropitant compared to that of ketamine would allow for rapid awakening with smooth recovery.

When opioids are contraindicated, maropitant could be part of an “opioid free anesthesia/analgesia” (OFA) anesthetic protocol [67,68].

Further studies with echocardiographic monitoring accompanied by serial blood gas analysis during longer surgical interventions, also through a higher nociceptive impact, would allow for a better evaluation of the cardiorespiratory effects of maropitant and of the entire anesthetic protocol.

## Figures and Tables

**Figure 1 animals-13-02388-f001:**
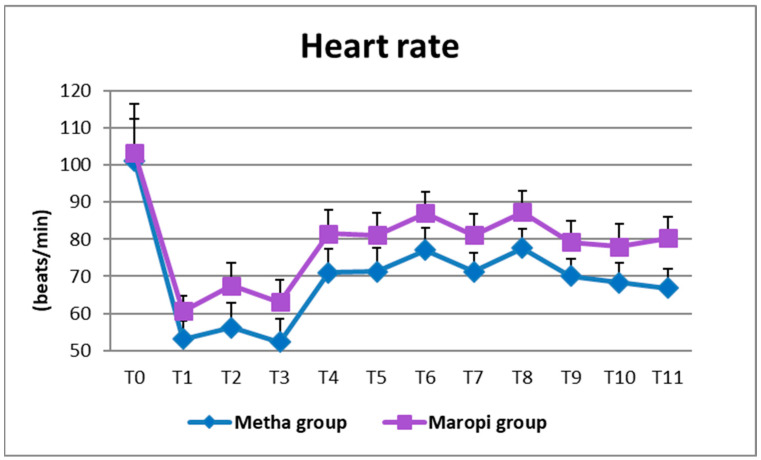
Mean ± SD of heart rate in dogs receiving maropitant (*n* = 20) or methadone (*n* = 20) from baseline values to skin suture (T0–T11).

**Figure 2 animals-13-02388-f002:**
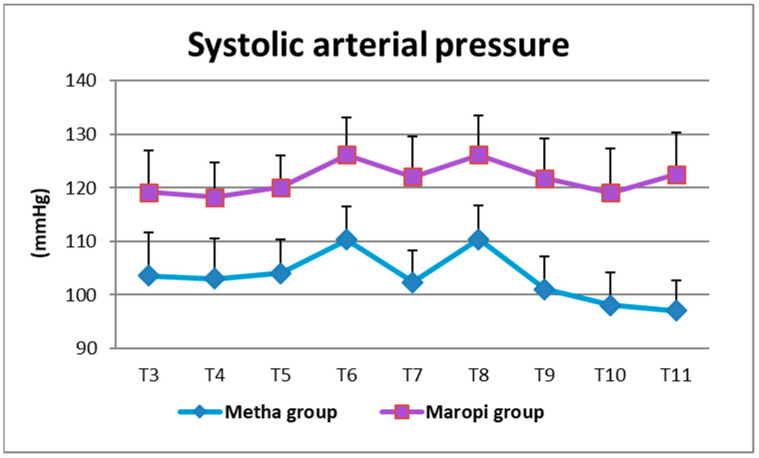
Mean ± SD of systolic arterial pressure in dogs receiving maropitant (*n* = 20) or methadone (*n* = 20) from immediately after intubation to skin suture (T3–T11).

**Figure 3 animals-13-02388-f003:**
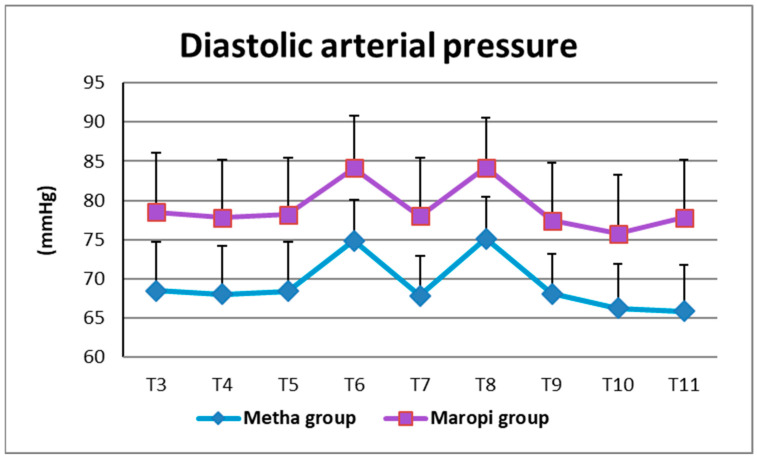
Mean ± SD of diastolic arterial pressure in dogs receiving maropitant (*n* = 20) or methadone (*n* = 20) from immediately after intubation to skin suture (T3–T11).

**Figure 4 animals-13-02388-f004:**
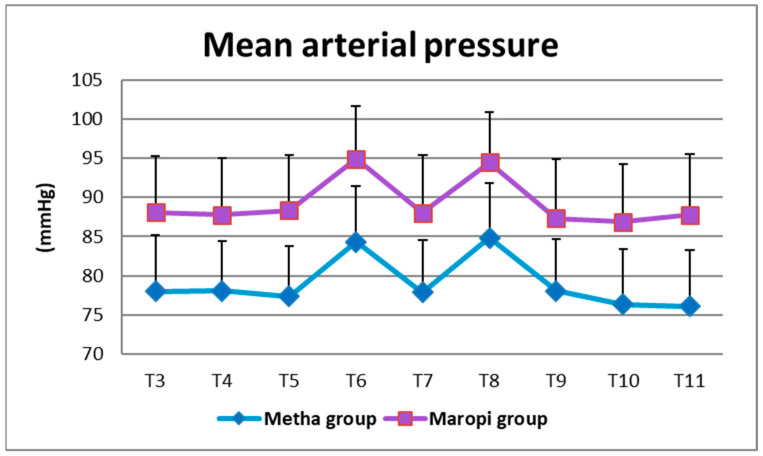
Mean ± SD of mean arterial pressure in dogs receiving maropitant (*n* = 20) or methadone (*n* = 20) from immediately after intubation to skin suture (T3–T11).

**Figure 5 animals-13-02388-f005:**
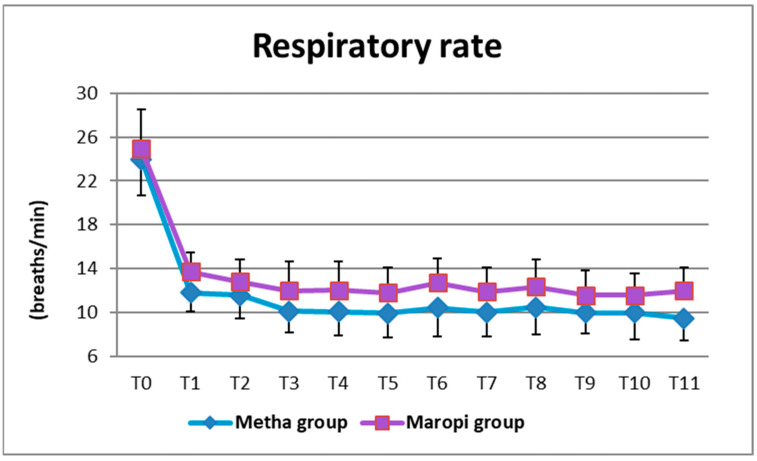
Mean ± SD of respiratory rate in dogs receiving maropitant (*n* = 20) or methadone (*n* = 20) from baseline values to skin suture (T0–T11).

**Figure 6 animals-13-02388-f006:**
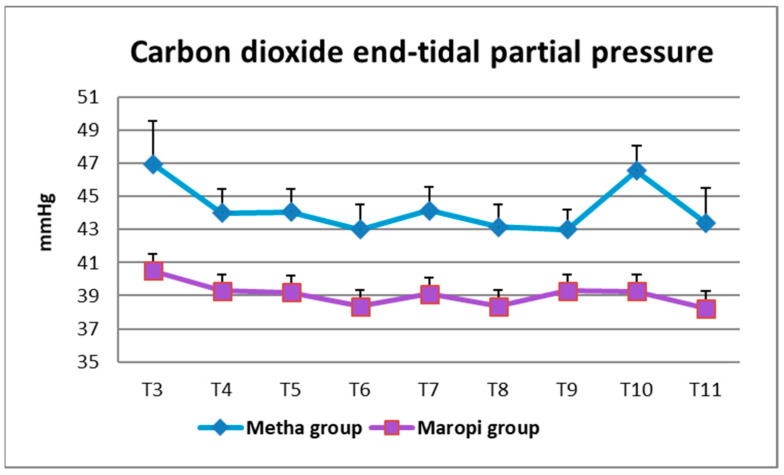
Mean ± SD of carbon dioxide end-tidal partial pressure in dogs receiving maropitant (*n* = 20) or methadone (*n* = 20) from immediately after intubation to skin suture (T3–T11).

**Figure 7 animals-13-02388-f007:**
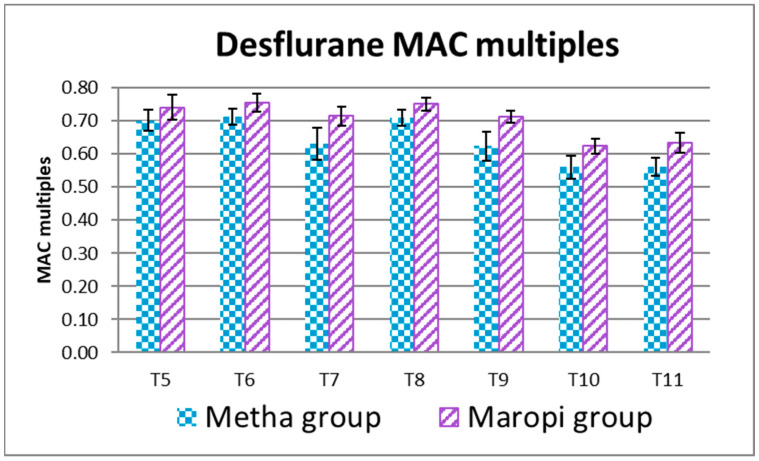
Mean ± SD of desflurane MAC multiples in dogs receiving maropitant (*n* = 20) or methadone (*n* = 20) from skin incision to skin suture (T5–T11).

**Figure 8 animals-13-02388-f008:**
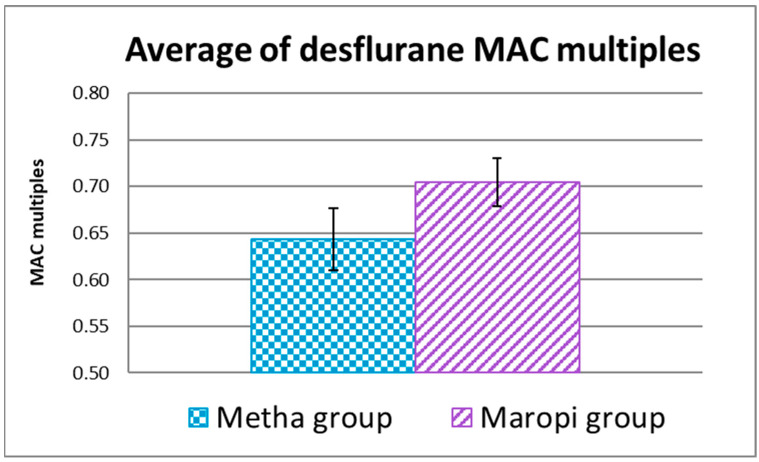
Average of EtDes expressed in MAC multiples referred to 7.64% as MAC in the dog [48] during the entire surgery (T5-T11) in dogs receiving maropitant (*n* = 20) or methadone (*n* = 20).

**Figure 9 animals-13-02388-f009:**
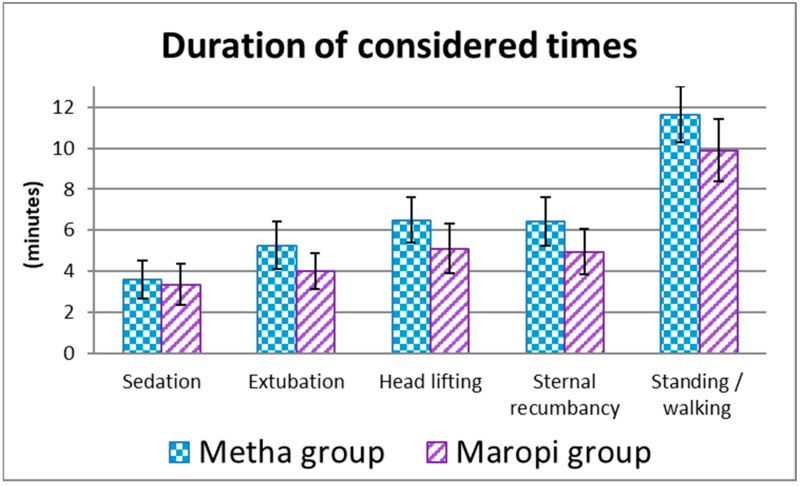
Mean ± SD of the duration of the considered times from premedication to recovery in dogs receiving maropitant (*n* = 20) or methadone (*n* = 20).

**Table 1 animals-13-02388-t001:** Time points of the collection of the variables from T0 to T11.

Evaluation Time Points	
T0	Baseline values
T1	Ten minutes after administration of premedication
T2	Shortly before induction
T3	Immediately after intubation
T4	Twenty minutes after EtDes 5.5%
T5	Skin incision
T6	Right ovarian ligament traction force
T7	Right ovarian ligament resection
T8	Left ovarian ligament traction force
T9	Left ovarian ligament resection
T10	Fascia suture
T11	Skin suture

**Table 2 animals-13-02388-t002:** Composite simple descriptive sedation score for dogs as described in the work of Grint 2009 [43] and Wagner 2017 [44].

Category	Expressions	Score
Spontaneous posture	Standing	0
	Tired but standing	1
	Lying but able to rise	2
	Lying but difficulty rising	3
	Unable to rise	4
Palpebral Reflex	Brisk	0
	Slow but with full corneal sweep	1
	Slow but with only partial corneal sweep	2
	Absent	3
Eye position	Central	0
	Rotated forwards/downwards but not obscured by third eyelid	1
	Rotated forwards/downwards and obscured by third eyelid	2
Jaw and tongue relaxation	Normal jaw tone/(strong gag reflex)	0
	Reduced tone, (but still moderate gag reflex)	1
	Much reduced tone, slight gag reflex	2
	Loss of jaw tone and no gag reflex	3
Response to noise (handclap)	Normal startle reaction (head turn towards noise/cringe)	0
	Reduced startle reaction (reduced head turn/minimal cringe)	1
	Minimal startle reaction	2
	Absent reaction	3
Resistance when laid into lateral recumbency	Much struggling, perhaps not allowing this positioning	0
	Some struggling, but allowing this positioning	1
	Minimal struggling/permissive	2
	No struggling	3
General appearance attitude	Excitable	0
	Awake and normal	1
	Tranquil	2
	Stuporous	3

**Table 3 animals-13-02388-t003:** Qualitative scoring system used for easy intubation.

Category	Expressions	Score
Excellent	Intubation successful in one attempt without physical reaction to intubation	0
Good	Intubation successful in one attempt with physical response to intubation	1
Satisfactory	Intubation successful after more than one attempt with or without physical response to intubation	2
Poor	Intubation impossible	3

**Table 4 animals-13-02388-t004:** Recovery scoring system as described by Hampton 2019 [45].

Factors	Assessments	Score
Struggling	None	0
	Transient, easily calmed by the investigator’s voice	1
	Prolonged (>1 min)	2
	Persistent (or requiring restraint)	3
Excitement	None	0
	Transient, easily calmed by the investigator’s voice	1
	Prolonged (>1 min)	2
	Persistent (or requiring restraint)	3
Paddling	None	0
	Transient, easily calmed by the investigator’s voice	1
	Prolonged (>1 min)	2
	Persistent (or requiring restraint)	3
Flailing	None	0
	Transient, easily calmed by the investigator’s voice	1
	Prolonged (>1 min)	2
	Persistent (or requiring restraint)	3
Vocalization	None	0
	Transient, easily calmed by the investigator’s voice	1
	Prolonged (>1 min)	2
	Persistent (or requiring restraint)	3
Administration of rescue drugs	Not given	0
	Given	3

**Table 5 animals-13-02388-t005:** Glasgow Composite Measure Pain Scale-Short Form (CMPS-SF) as described by Reid 2007 [46].

**S**HORT **F**ORM OF THE **G**LASGOW **C**OMPOSITE **P**AIN **S**CALE			
Dog’s name _______________________________________________			
Hospital Number _________	Date / /	Time __________	
Surgery Yes/No (delete as appropriate)			
Procedure or Condition **___________________________________________**			
In the sections below please circle the appropriate score in each list and sum these to give the total score			
**A.** **Look at dog in kennel.**			
(i)		(ii)	
Quiet	0	Ignoring any wound or painful area	0
Crying or whimpering	1	Looking at wound or painful area	1
Groaning	2	Licking wound or painful area	2
Screaming	3	Rubbing wound or painful area	3
		Chewing wound or painful area	4
**__________________________________________________________________________**			
In the case of spinal, pelvic or multiple limb fractures, or where assistance is required to aid locomotion do not carry out section B and proceed to C Please tick if this is the case ◻ then proceed to C			
**B.** **Put lead on dog and lead out of the kennel.**		**C.** **If it has a wound or painful area including abdomen, apply gentle pressure 2 inches round the site.**	
**When the dog rises/walks is it?**		**Does it?**	
(iii)		(iv)	
Normal	0	Do nothing	0
Lame	1	Look round	1
Slow or reluctant	2	Flinch	2
Stiff	3	Growl or guard area	3
It refuses to move	4	Snap	4
		Cry	5
**__________________________________________________________________________**			
**D.** **Overall.**			
**Is the dog?**		**Is the dog?**	
(v)		(vi)	
Happy and content or happy and bouncy	0	Comfortable	0
Quiet	1	Unsettled	1
Indifferent or non-responsive to surroundings	2	Restless	2
Nervous or anxious or fearful	3	Hunched or tense	3
Depressed or non-responsive to stimulation	4	Rigid	4
**© University of Glasgow**		**Total Score (i + ii + iii + iv + v + vi) = _______**	

**Table 6 animals-13-02388-t006:** Mean ± SD tidal volume (VT) and minute volume (VM) in dogs receiving maropitant (*n* = 20) or methadone (*n* = 20).

	Maropi Group		Metha Group	
	VT (mL)	VM (Lt)	VT (mL)	VM (Lt)
T3	192.13 ± 31.78	1.93 ± 2.47	195.13 ± 34.81	1.90 ± 3.44
T4	192.72 ± 30.74	1.90 ± 3.91	195.72 ± 33.77	1.87 ± 2.88
T5	193.08 ± 29.63	1.88 ± 2.15	196.08 ± 32.66	1.85 ± 3.12
T6	191.75 ± 26.08	1.95 ± 2.95	194.75 ± 29.11	1.92 ± 2.92
T7	193.23 ± 31.78	1.90 ± 2.14	196.23 ± 34.81	1.87 ± 3.11
T8	191.00 ± 37.14	1.97 ± 3.97	194.00 ± 30.17	1.94 ± 2.94
T9	191.65 ± 31.78	1.91 ± 3.12	195.65 ± 34.81	1.88 ± 3.09
T10	193.17 ± 30.78	1.89 ± 2.79	196.17 ± 34.81	1.86 ± 2.77
T11	194.13 ± 24.34	1.88 ± 2.42	197.13 ± 27.37	1.86 ± 3.40

## Data Availability

Not applicable.

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
