# Peer review of "Cardiorespiratory Effects and Desflurane Requirement in Dogs Undergoing Ovariectomy after Administration Maropitant or Methadone"

_animals, 2023, doi:10.3390/ani13142388_

Round 1

Reviewer 1 Report

Cardiorespiratory effects and desflurane requirement in dogs undergoing ovariectomy premedicated with dexmedetomidine and induced with propofol, after administration of maropitant or methadone

I appreciate the opportunity to revise the present study. Opioid-free anesthesia/analgesia is a current trend aiming to decrease or prevent the adverse effects that opioids might cause. The present study provides interesting and significant findings where maropitant has a similar analgesic effect as methadone, which is a protocol that could be used in further research.

Line 2. By reading the entire article, methods and, results, although the premedication and induction are important, the effect of these two drugs (dexmedetomidine and propofol) were not really part of the evaluated variables, in contrast to cardiorespiratory and requirement values, am I right? I would suggest considering limiting the title to the effects that desflurane, in combination with maropitant or methadone, had on the dogs undergoing ovariectomy (without mentioning propofol and dexmedetomidine).

Response:

Lines 12-16: I suggest changing the order of the sentences at the beginning of the simple summary and adding the importance of pain management during a surgical procedure. A recommendation would be: “Analgesic drugs in the anesthetic protocol can reduce the requirement of other drugs, particularly inhalant agents, and are essential for pain management during a surgical procedure. Opioids such as methadone are the most potent and most used analgesic drugs in anesthetic protocols, but they have several dose-dependent adverse effects. Some drugs other than opioids also have analgesic effects. Maropitant….”

Response:

Line 17: It is a little confusing reading the title, the objective written in the simple summary, the abstract, and the introduction, because they have different elements and differ among them. In the title, it says that the authors are assessing the cardiorespiratory effects and requirements of desflurane, but in the objective, there is no mention of the cardiorespiratory part, only the requirements of desflurane. My recommendation is to include in the aim of the study the cardiorespiratory effects and analgesic properties of maropitant and methadone combined with desflurane in dogs undergoing ovariectomy. Also, homologate the aim presented in the simple summary, the abstract, and the introduction.

Response:

 Lines 19-20: Here, the authors state that cardiorespiratory functions were maintained. However, there is no mention in the study’s objective that cardiorespiratory function was assessed. As per my previous comment, I suggest including this element in the objective of the study and clearly mention which parameters were evaluated in dogs (e.g., heart rate, respiratory rate, rectal temperature, SAP, DAP, MAP, BT, VT, VM, PetcO2, etc.).

Response:

 Line 30. Please, indicate more details about the experimental characteristics. For example, the administration route of the drugs, the breed of the dogs, which evaluation times were used, etc.

Response:

Lines 31- 33: An important parameter that could be included in the conclusions is the reduction of desflurane MAC when using maropitant or methadone.

Response:

Line 48: The authors could mention the concept of “multimodal analgesia” and also mention that opioid use is often restricted in companion animals (only 13% of dogs receive an opioid during a surgical procedure) due to legal aspects and the presentation of adverse effects. These articles might be helpful: https://doi.org/10.1111/j.1467-2987.2004.00175.x, https://doi.org/10.1016/j.tvjl.2018.06.003, http://dx.doi.org/10.31893/jabb.23009

Response:

Line 50: Please, amend the in-text citation style for Ingvast-Larsson 2010 and other citations throughout the text.

Response:

Lines 79-96: An important factor to mention in the Introduction is that due to the adverse effects that opioids might have, the use of other drugs with similar analgesic properties as opioids is a field of research in companion animals’ anesthesia and is part of the novelty of the anesthetic protocol suggested by the authors. The authors could refer to these studies: Miranda Cortes (2020) Cardiorespiratory effects of epidurally administered ketamine or lidocaine in dogs undergoing ovariohysterectomy surgery: a comparative study and https://doi.org/10.1371/journal.pone.0223697. 

Response:

 Lines 97-100: Clearly state that this was the objective of the study and specify which cardiorespiratory parameters were assessed in dogs.

Response:

Lines 101-104: If possible, add the hypothesis regarding the cardiorespiratory effect of both drugs (since it is part of the aim of the study).

Response:

Line 105: In general, for the Materials and Methods section, consider dividing the sections according to what the authors are presenting. For example: 2.1. Ethical statement; 2.2. Study subjects; 2.3. Anesthetic and analgesic protocol/management; 2.4. Surgical procedure; 2.5. Cardiorespiratory parameters; 2.6. Evaluation times/Time points of evaluation; 2.7. Postoperative analgesic management, etc. This would help to organize the information written in this section and give a little more order in the description of the methods. As it is now, in some parts it is confusing to understand which variables were recorded at which time points and the description of these times starts at Basal-T3, T3-T11, then goes back to T4.

Response:

 Lines 114-118: Mention if all types of breeds were included in the study.

Response:

Lines 126: Specify what you refer to as “basal measurement”. Was this 12 hours before the surgical procedure? Immediately before the procedure? Immediately before administrating premedication?

Response:

Line 135: Use the abbreviation of “heart rate” accordingly. In line 239 as well.

Response:

Lines 143-148: Heart rate was no longer assessed from T3 to T11? If so, please, include it in the evaluated parameters. Also, I consider that Table 1 needs to be cited the first time the authors are mentioned the time points of the collection of the variables.

Response:

Tables 2, 3 ,and 4: Amend the in-text citation style in the table title. In Table 4 correct “Jaw e tongue relaxation”. Also, as a general comment for all Tables and Figures, each requires a title. For example, “Figure 1. Mean ± standard deviation (SD) of heart rate in dogs receiving maropitant (n=20) or methadone (n=20) from ten minutes after administration of premedication to skin suture (T1-T11). It is not entirely necessary to include in the Figure title that Maropi or Metha groups were lower/higher. This information is already provided in the main text.

Response:

 Line 290: Correct “Figures 8” with “Figure 8”.

Response:

Lines 296-297: What about the scores of the CMPS-SF? It is clear that dogs did not require rescue analgesia but it would be interesting to include the average score of the subjects in each experimental group.

Response:

Line 306-307: I recommend starting the discussion by mentioning the main findings when comparing Maropi and Metha groups.

Response:

Lines 322-324: Here, the authors could enhance the relevance of their study by mentioning that the properties of maropitant, combined with desflurane, do not only provide an acceptable analgesia protocol comparable to methadone but also helps to reduce patient recovery times and the consequences that a prolonged recovery might represent for dogs (https://doi.org/10.1371/journal.pone.0140734, Hay Kraus, B. L. (2017). Spotlight on the perioperative use of maropitant citrate. Veterinary Medicine: Research and Reports, 41-51).

Response:

Lines 340: It would be interesting to discuss how the mechanism of action of maropitant can be considered as a benefit when compared to opioids. This would help to understand all the cardiorespiratory findings and the lack of a marked cardiorespiratory depression when compared to methadone.   

Response:

Lines 373, 380, 386, and so on: Amend the citation style.

Response:

Line 403: When referring to a time point of evaluation (skin suture), please, use the designated abbreviation for this time.

Response:

Lines 411-415: I think that the comparable analgesic effect of methadone and maropitant is an element that needs to be highlighted and mentioned together with what the authors wrote in lines 421-422 about maropitant being a potential option to administer opioid-free anesthesia/analgesia.

Response:

Lines 417-420: While this sentence might be true, the authors of the present article did not assess or compare the effect of premedication with an alpha 2 agonist and phenothiazine to reach this conclusion. I would suggest moving this paragraph to the discussion.

Response:

Author Response

Dear Reviewer.

Answers and changes referred to your comments are in red.

Exception has been done for the chapters of materials and methods and discussion-conclusions, which in accordance with your suggestions, have been considerably modified and corrected in the succession of their concepts expressed. For this reason and for easier reading, the texts are in black.

The corrections relating to the other reviewer are in light blue.

Comments and Suggestions for Authors

Cardiorespiratory effects and desflurane requirement in dogs undergoing ovariectomy premedicated with dexmedetomidine and induced with propofol, after administration of maropitant or methadone

I appreciate the opportunity to revise the present study. Opioid-free anesthesia/analgesia is a current trend aiming to decrease or prevent the adverse effects that opioids might cause. The present study provides interesting and significant findings where maropitant has a similar analgesic effect as methadone, which is a protocol that could be used in further research.

Line 2:

By reading the entire article, methods and, results, although the premedication and induction are important, the effect of these two drugs (dexmedetomidine and propofol) were not really part of the evaluated variables, in contrast to cardiorespiratory and requirement values, am I right? I would suggest considering limiting the title to the effects that desflurane, in combination with maropitant or methadone, had on the dogs undergoing ovariectomy (without mentioning propofol and dexmedetomidine).

Response: The title has been adjusted accordingly.

Lines 12-16:

I suggest changing the order of the sentences at the beginning of the simple summary and adding the importance of pain management during a surgical procedure. A recommendation would be: “Analgesic drugs in the anesthetic protocol can reduce the requirement of other drugs, particularly inhalant agents, and are essential for pain management during a surgical procedure. Opioids such as methadone are the most potent and most used analgesic drugs in anesthetic protocols, but they have several dose-dependent adverse effects. Some drugs other than opioids also have analgesic effects. Maropitant….”

Response: The order has been changed, it sounds much better.

Line 17:

It is a little confusing reading the title, the objective written in the simple summary, the abstract, and the introduction, because they have different elements and differ among them. In the title, it says that the authors are assessing the cardiorespiratory effects and requirements of desflurane, but in the objective, there is no mention of the cardiorespiratory part, only the requirements of desflurane. My recommendation is to include in the aim of the study the cardiorespiratory effects and analgesic properties of maropitant and methadone combined with desflurane in dogs undergoing ovariectomy. Also, homologate the aim presented in the simple summary, the abstract, and the introduction.

Response: The text has been changed consequently. The aim has been adjusted and homologated. It was really everywhere different and incomplete. Thank you.

Lines 19-20:

Here, the authors state that cardiorespiratory functions were maintained. However, there is no mention in the study’s objective that cardiorespiratory function was assessed. As per my previous comment, I suggest including this element in the objective of the study and clearly mention which parameters were evaluated in dogs (e.g., heart rate, respiratory rate, rectal temperature, SAP, DAP, MAP, BT, VT, VM, PetcO2, etc.).

Response: Yes, for mistake there were no mention. The parameters have been now included.

Line 30:

Please, indicate more details about the experimental characteristics. For example, the administration route of the drugs, the breed of the dogs, which evaluation times were used, etc.

Response: Some more details have been indicated.

Lines 31- 33:

An important parameter that could be included in the conclusions is the reduction of desflurane MAC when using maropitant or methadone.

Response: Since we did not calculate the MAC of our trial, we are only allowed to discuss the reduction in requirement, and this was stated in the results.

Line 48:

The authors could mention the concept of “multimodal analgesia” and also mention that opioid use is often restricted in companion animals (only 13% of dogs receive an opioid during a surgical procedure) due to legal aspects and the presentation of adverse effects. These articles might be helpful:

https://doi.org/10.1111/j.1467-2987.2004.00175.x;

https://doi.org/10.1016/j.tvjl.2018.06.003;

http://dx.doi.org/10.31893/jabb.23009.

Response: The concept has been mentioned together with its bibliography.

Line 50:

Please, amend the in-text citation style for Ingvast-Larsson 2010 and other citations throughout the text.

Response: Amendment have been done. Thank you.

Lines 79-96:

An important factor to mention in the Introduction is that due to the adverse effects that opioids might have, the use of other drugs with similar analgesic properties as opioids is a field of research in companion animals’ anesthesia and is part of the novelty of the anesthetic protocol suggested by the authors.

The authors could refer to these studies:

  • Miranda Cortes (2020) Cardiorespiratory effects of epidurally administered ketamine or lidocaine in dogs undergoing ovariohysterectomy surgery: a comparative study;
  • and https://doi.org/10.1371/journal.pone.0223697.

Response: The concept has been mentioned together with its bibliography.

Lines 97-100:

Clearly state that this was the objective of the study and specify which cardiorespiratory parameters were assessed in dogs.

Response: Objective has been stated and cardiorespiratory parameters have been specified.

Lines 101-104:

If possible, add the hypothesis regarding the cardiorespiratory effect of both drugs (since it is part of the aim of the study).

Response: The mention of the cardiorespiratory effects has been added in the hypothesis, even if without specifying the functions in order not to add too much weight to the text.

Line 105:

In general, for the Materials and Methods section, consider dividing the sections according to what the authors are presenting. For example:

2.1. Ethical statement;

2.2. Study subjects;

2.6. Evaluation times/Time points of evaluation;

2.3. Anesthetic and analgesic protocol/management;

2.4. Surgical procedure;

2.5. Cardiorespiratory parameters;

2.7. Postoperative analgesic management, etc.

This would help to organize the information written in this section and give a little more order in the description of the methods. As it is now, in some parts it is confusing to understand which variables were recorded at which time points and the description of these times starts at Basal-T3, T3-T11, then goes back to T4.

Response: The section has been redrafted and divided accordingly.

Lines 114-118:

Mention if all types of breeds were included in the study.

Response: The dog breeds have been mentioned. The bitches were all mixed breeds, because they came from different municipal kennels and were part of a sterilization program in shelter medicine.

Lines 126:

Specify what you refer to as “basal measurement”. Was this 12hours before the surgical procedure? Immediately before the procedure? Immediately before administrating premedication?

Response: Baseline measurement has been obtained shortly before IV access (T0) and has been better specified in the text.

Line 135:

Use the abbreviation of “heart rate” accordingly. In line 239 as well.

Response: It has been preferred not to start the sentence with an abbreviation.

Lines 143-148:

Heart rate was no longer assessed from T3 to T11? If so, please, include it in the evaluated parameters. Also, I consider that Table 1 needs to be cited the first time the authors are mentioned the time points of the collection of the variables.

Response: The corrections have been inserted. Thank you.

Tables 2, 3 and 4:

Amend the in-text citation style in the table title.

In Table 4 correct “Jaw e tongue relaxation”.

Also, as a general comment for all Tables and Figures, each requires a title. For example, “Figure 1. Mean ± standard deviation (SD) of heart rate in dogs receiving maropitant (n=20) or methadone (n=20) from ten minutes after administration of premedication to skin suture (T1-T11). It is not entirely necessary to include in the Figure title that Maropi or Metha groups were lower/higher. This information is already provided in the main text.

Response: The in-text citation style in the tables 2, 3 and 4 title have been corrected.

In table 4 “Jaw e tongue relaxation” has been corrected with “Jaw & tongue relaxation”.

The titles of the figures have been changed and the information already provided in the main text have been removed as suggested.

Line 290:

Correct “Figures 8” with “Figure 8”.

Response: “Figures 8” has been corrected with “Figure 8”.

Lines 296-297:

What about the scores of the CMPS-SF? It is clear that dogs did not require rescue analgesia but it would be interesting to include the average score of the subjects in each experimental group.

Response: Average scores have been included.

Line 306-307:

I recommend starting the discussion by mentioning the main findings when comparing Maropi and Metha groups.

Response: Some of the main findings have been positioned at the start of the discussion.

Lines 322-324:

Here, the authors could enhance the relevance of their study by mentioning that the properties of maropitant, combined with desflurane, do not only provide an acceptable analgesia protocol comparable to methadone but also helps to reduce patient recovery times and the consequences that a prolonged recovery might represent for dogs (https://doi.org/10.1371/journal.pone.0140734, Hay Kraus, B. L. (2017). Spotlight on the perioperative use of maropitant citrate. Veterinary Medicine: Research and Reports, 41-51).

Response: The concept of the properties of maropitant combined with desflurane has been inserted near the end of the discussion, after having rearranged the entire chapter as suggested.

Lines 340:

It would be interesting to discuss how the mechanism of action of maropitant can be considered as a benefit when compared to opioids. This would help to understand all the cardiorespiratory findings and the lack of a marked cardiorespiratory depression when compared to methadone.

Response: The concept of the mechanism of action of maropitant has been inserted in the start of the discussion, after having rearranged the entire chapter as suggested.

Lines 373, 380, 386, and so on:

Amend the citation style.

Response: Citation style has been amended.

Line 403:

When referring to a time point of evaluation (skin suture), please, use the designated abbreviation for this time.

Response: Time point of evaluation “skin suture” (T11) designated has been added. Thank you.

Lines 411-415:

I think that the comparable analgesic effect of methadone and maropitant is an element that needs to be highlighted and mentioned together with what the authors wrote in lines 421-422 about maropitant being a potential option to administer opioid-free anesthesia/analgesia.

Response: The comparable analgesic effect of methadone and maropitant has been mentioned.

Lines 417-420:

While this sentence might be true, the authors of the present article did not assess or compare the effect of premedication with an alpha 2 agonist and phenothiazine to reach this conclusion. I would suggest moving this paragraph to the discussion.

Response: The paragraph has been moved to the discussion.

Reviewer 2 Report

Congratulations to the authors on this very relevant publication for veterinary surgeons in practice. 

The introduction gives a good background. The study design is very clear and rigid; being double-blind, using the same operator, only using healthy animals, etc.  The results are clearly described, the discussion puts the findings in a broader field. 

I have made some suggestions for improvement in the attached pdf, related to making the tables more readable, some typos and some wording suggestions. 

One additional strength could be to add fields in which you think additional research is needed. 

Author Response

Dear Reviewer.

Answers and changes referred to your comments are in light blue.Inizio modulo

Comments and Suggestions for Authors

Congratulations to the authors on this very relevant publication for veterinary surgeons in practice. 

The introduction gives a good background. The study design is very clear and rigid; being double-blind, using the same operator, only using healthy animals, etc.  The results are clearly described, the discussion puts the findings in a broader field. 

I have made some suggestions for improvement in the attached pdf, related to making the tables more readable, some typos and some wording suggestions. 

Response: All suggestions has been taken into consideration and tried to be addressed. Thank you.

One additional strength could be to add fields in which you think additional research is needed.

Response: Fields where further research is needed have been added. Thank you.

L 12

This is a minor comment, but I would use instead of 'drug' the term 'medicine'. The term drug is wider, has a more negative commutation and also includes illegal substances. The term medicine is more positive and patient-focused aspects of therapeutic substances.

Response: I find it difficult to replace the term "drug" in the text. Drug is often used in the literature. For example, in Table 4 which we took from Hampton 2019.

L22

Is there a specific reason for only advising to use maropitant as analgesic for dogs with reduced cardiorespiratory or hypotension and not for all dogs?

Response: Thanks for the remark. In the text it has been better specified.

L25

While agreeing this is the best anesthesia protocal, it is not the only one. Some vets are still using injection anaesthesia (not ideal). So I would rephrase the sentence to 'a balanced anesthesia protocol such as using analgesics along with an inhalation anaesthetic'.

Response: The sentence has been corrected.

L28

Please use past simple tense throughout the publication.

Response: Past simple tense has been used throughout the publication.

L70

Please define MAC/Minimum alveolar concentration

Response: MAC has been defined.

L 194

Maybe say 'was available to be provided' as no rescue tx was necessary.

Response: The sentence has been rephrased.

L243

Looking at these graphs in a black/white print you can not differentiate between the two lines.

So I suggest to make the lines different e.g. with dots or with scares, etc.

Same comment for the other graphs.

Response: The lines of all graphs have been changed to be recognizable even in black/white printing.

L 257

Copy mistake - Diastolic

Copy mistake - Mean arterial pressure

Response: On advice of the other reviewer, the captions of all the figures have been entirely changed.

L 263

Would it not be more logical to put Fig 5 after Fig 1?

Response: First all the cardiovascular variables were described and then the respiratory ones.

L 320

This sentence is a bit unclear to me.

Response: The sentence has been rephrased. We wanted to express the concept that for the respect of animal welfare, the presence of a drug with some analgesic effect was ensured even in dogs that did not receive opioids. The analgesic effect of maropitant was not jet certain at the beginning of the trial.

Round 2

Reviewer 1 Report

The authors have made extensive changes and adjustments according to my feedback.

I have no additional comments.

The article must be published.